# Anterior chamber depth in mice is controlled by several quantitative trait loci

Demelza R. Larson[1]*, Allysa J. Kimber[1], Kacie J. Meyer[2], Michael G. Anderson[2,3,4]

**1** Department of Biology, College of Saint Benedict & Saint John's University, Collegeville, Minnesota, United States of America, **2** Department of Molecular Physiology and Biophysics, The University of Iowa, Iowa City, Iowa, United States of America, **3** Department of Ophthalmology and Visual Sciences, The University of Iowa, Iowa City, Iowa, United States of America, **4** Center for the Prevention and Treatment of Visual Loss, Iowa City VA Health Care System, Iowa City, Iowa, United States of America

* drlarson@csbsju.edu

## Abstract

Anterior chamber depth (ACD) is a quantitative trait associated with primary angle closure glaucoma (PACG). Although ACD is highly heritable, known genetic variations explain a small fraction of the phenotypic variability. The purpose of this study was to identify additional ACD-influencing loci using strains of mice. Cohorts of 86 N2 and 111 F2 mice were generated from crosses between recombinant inbred BXD24/TyJ and wild-derived CAST/EiJ mice. Using anterior chamber optical coherence tomography, mice were phenotyped at 10–12 weeks of age, genotyped based on 93 genome-wide SNPs, and subjected to quantitative trait locus (QTL) analysis. In an analysis of ACD among all mice, six loci passed the significance threshold of $p = 0.05$ and persisted after multiple regression analysis. These were on chromosomes 6, 7, 11, 12, 15 and 17 (named *Acdq6*, *Acdq7*, *Acdq11*, *Acdq12*, *Acdq15*, and *Acdq17*, respectively). Our findings demonstrate a quantitative multi-genic pattern of ACD inheritance in mice and identify six previously unrecognized ACD-influencing loci. We have taken a unique approach to studying the anterior chamber depth phenotype by using mice as genetic tool to examine this continuously distributed trait.

## Introduction

Anterior chamber depth (ACD) is defined as the distance between the innermost layer of the cornea and the anterior face of the lens in the anterior chamber of the eye. In human populations, ACD is normally distributed, suggesting it is a quantitative trait with a complex inheritance pattern. In populations where it has been measured, ACD confers heritability estimates up to 90%, indicating that differences in genetic factors control the majority of the phenotypic variability [1–4]. Despite this high heritability, currently known genetic variations explain only a small fraction of this phenotypic variability.

In humans, eyes with shallow anterior chambers tend to have a higher incidence of being hypermetropic and are at a greater risk for developing primary angle closure glaucoma (PACG) than eyes with average or deep anterior chambers [5]. Shallow ACD is associated with narrow openings of the drainage structures of the anterior chamber of the eye and causes a subsequent rise in intraocular pressure that is a hallmark of many forms of glaucoma. It has

**Data Availability Statement:** All relevant data are within the paper and its Supporting information files, except for the file containing the mouse genotype/phenotype data. That information has

been deposited on figshare (10.6084/m9.figshare.23664273).

**Funding:** The author(s) received no specific funding for this work.

**Competing interests:** The authors have declared that no competing interests exist.

been demonstrated that at least one of the genes that control ACD (*ABCC5*) is also associated with PACG [6]. This suggests that identifying the genetic elements underlying ACD, an important endophenotype of PACG, could be a useful strategy for also uncovering the genetic factors contributing to PACG itself.

An effective way of identifying genetic factors underlying complex traits is to use quantitative trait locus (QTL) analysis with mice [7]. Eyes from both mice and humans are structurally and functionally similar and are therefore likely to be influenced by similar genetic pathways. Like humans, there is a natural variability in ACD in different strains of inbred mice, as we show here. In this study, we have used phenotype-driven mouse genetics to map QTL that influence ACD. We report results from crosses between two inbred strains of mice that identify at least 6 previously unknown ACD influencing loci.

## Materials and methods

### Experimental animals

BXD24/TyJ-*Cep290^{rd16}*/J (abbreviated as BXD24b throughout) and CAST/EiJ (abbreviated as CAST throughout) mice were obtained from The Jackson Laboratory and subsequently housed and bred at the University of Iowa Research Animal Facility. BXD24b is a recombinant inbred strain of mice derived from C57BL/6J and DBA/2J progenitors [8] and contains a spontaneous mutation within the *Cep290* gene [9]. CAST is an inbred strain originally derived from wild mice trapped in a grain warehouse in Thailand [10]. Studies have shown substantial genetic differences in CAST mice compared to both B6 and D2 mice, which make that strain a useful genetic tool for QTL studies like this one [11]. (BXD24b X CAST) F1 mice were either backcrossed with BXD24b mice to produce a population of (BXD24b X CAST) N2 mice, or intercrossed to produce F2 mice. The study on anterior chamber depth reported here was an accompaniment to a broader ongoing study that seeks to identify genetic modifiers of the recessive *Cep290^{rd16}* mutation; therefore, all N2 and F2 mice available for the study were pre-selected, by PCR-based genotyping at weaning, for homozygosity of *Cep290^{rd16}*. A subset of these mice were also used as part of a supplementary study on central corneal thickness (CCT) [12]. All animals were treated in accordance with the ARVO Statement for the Use of Animals in Ophthalmic and Vision Research. All experimental protocols were approved by the Animal Care and Use Committee of the University of Iowa.

### Mouse genotyping

Genomic DNA was isolated from ear tissue of each mouse. Genome-wide genotyping of genomic DNA was performed at the University of Iowa using Fluidigm technology, following the manufacturer's instructions, and a panel of 93 (Fluidigm Corporation, South San Francisco, CA) assays for single nucleotide polymorphisms that differentiate BXD24b from CAST alleles [13]. The average spacing between markers was 16 cM (Fig 1, [12]). For SNP assays, DNA was simultaneously PCR amplified from each sample using a multiplex PCR kit (Qiagen), with specific target-amplification (STA) primers and locus-specific primers (LSP). Pre-amplified DNA samples were diluted 1:10 and then combined with allele-specific primers (ASP), LSP, and the required Fluidigm buffers and reagents, and loaded into the integrated fluidic circuits for SNP genotyping. Genotyping calls were made using the SNP Genotyping Analysis Software v.3.0.2 and Fluidigm Data Collection Software v.3.0.2. All primer sequences are available upon request [12].

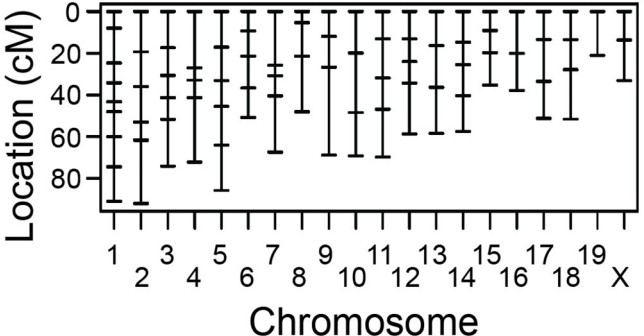

**Fig 1. Distribution of polymorphic markers across the mouse genome.** The horizontal dashes along each chromosome represent one marker. The mean distance between markers is 16 cM [12].

## ACD phenotyping

All measurements were recorded from 10–12 week-old mice, a time at which the adult eye has fully developed [14], using ocular anterior segment images obtained with a Bioptigen spectral domain optical coherence tomographer (SD-OCT; Bioptigen, Inc., USA). SD-OCT is a non-contact, non-invasive and painless procedure; however, anesthesia was required to keep the mice perfectly still during imaging. The animals were injected with a standard mixture of keta-mine/xylazine (intraperitoneal injection of 100 mg ketamine + 10 mg xylazine / kg body weight; Ketaset®, Fort Dodge Animal Health, Fort Dodge, IA; AnaSed®, Lloyd Laboratories, Shenandoah, IA). During induction of, and recovery from, anesthesia, the mice were provided supplemental indirect warmth by a heating pad; the mice were not euthanized at the time of this study. Immediately following anesthesia, eyes were hydrated with balanced salt solution (BSS; Alcon Laboratories, Fort Worth, TX). A 12-mm telecentric bore with a reference arm position of 1048 was used to image the anterior segment of each eye. The bore was positioned such that the pupil of the eye was centered in the volume intensity projection. Scan parameters were as follows: radial volume scans 2.0 mm in diameter, 1000 A-scans/B-scan, 100 B-scans/volume, 1 frame/B-scan, and 1 volume. Using the ImageJ software, anterior chamber depth was measured on saved SD-OCT JPEG images. Measurements were taken from the innermost layer of the cornea to the anterior face of the lens. Mice were included in the analysis if the standard deviation (SD) between the right and left eyes was less than 20 μm, and if both eyes were free from opacity [12]. The spreadsheets containing the genotype and phenotype data for all the mice used for the QTL statistical analysis (described below) can be found on figshare (10.6084/m9.figshare.23664273).

## Statistical analysis

Statistical comparisons of ACD between parental strains of mice (BXD24b and CAST) were calculated using an unpaired two-tailed Student's *t*-test. QTL analysis was performed with R/qtl, using the N2 and F2 datasets separately and in combination [15–17]. The genome-wide scan (scanone) and two dimensional genome-wide scan (scantwo) were conducted as previously described [12, 18]. The significance thresholds for the genome-wide scan were determined by performing traditional permutation testing, using 1,000 permutations. Loci with LOD scores above the $p = 0.05$ threshold were considered significant QTL, loci with LOD scores above the $p = 0.63$ threshold were considered suggestive QTL [19], and loci with LOD scores above the $p = 0.1$ threshold were considered highly suggestive. For the two-dimensional

genome-wide scan, significance thresholds were determined empirically by permutation testing, using 1,000 permutations [12].

The validity of a multiple QTL model was tested by performing multiple regression analysis. Phenotypic variance was estimated and the full model was statistically compared to reduced models in which one QTL was dropped. The analysis follows the formula: $LOD(QTL_1) = \log10 [Pr(data|QTL_1, QTL_2, ..., QTL_k) / Pr(data|QTL_2, ..., QTL_k)]$ [15]. If the probability of the data with all "k" QTL is close to the last "k-1" QTL, then the LOD score is low and support for $QTL_1$ within the model is decreased. If the data are more probable when $QTL_1$ is included in the model, then the LOD score is large and support for $QTL_1$ within the model is increased. To determine the level of support for the LOD scores resulting from the multiple regression analysis, they were compared to the significance thresholds from the one-dimensional genome-wide scan [12].

To compare differences of allelic effects at a single SNP marker, an unpaired two-tailed Student's *t*-test was used. Differences in ACD were considered significant if the *p*-values were less than 0.05 and the level of confidence was >95% after correction for multiple comparisons. Unless otherwise stated, all ACD values are reported based on the number of mice and are expressed as an average ± SD.

## Bioinformatics analysis

The 95% Bayes credible intervals for significant QTL were calculated using R/qtl. The protein-coding genes and noncoding RNA genes within the Bayes credible intervals for QTL were identified using the Mouse Genome Informatics (MGI) "Genes & Markers Query" (http://www.informatics.jax.org/marker). The interrogated cM positions for the genes of interest spanned the following intervals: for *Acdq6*, Chr 6 from 2.8cM–45.3cM; for *Acdq7*, Chr 7 from 36.4cM–78.9cM; for *Acdq11*, Chr 11 from 8.6cM–26.1cM; for *Acdq12*, Chr 12 from 31.9cM—56.9cM; *Acdq15*, Chr 15 from 20.3cM—35.3cM; *Acdq17*, Chr 17 from 25.5cM—55.5cM. The genome base-position coordinates provided in supplementary tables are from assembly GRCm39/mm39. The resulting MGI gene lists were then filtered such that genes were included as candidates underlying the QTL only if they 1) contained one or more potential DNA base-pair changes that affect an encoded protein and 2) were expressed in mouse ocular tissue (see S1–S6 Tables).

To identify genes with potentially important DNA base-pair changes within the Bayes credible intervals, the chromosomal regions used for the MGI query (converted to GRCm38/mm10 assembly coordinates using the Lift Genome Annotations website: https://genome.ucsc.edu/cgi-bin/hgLiftOver) were examined using the Wellcome Trust Sanger Institute's Mouse Genomes Project "SNP and Indel Query" tool (http://www.sanger.ac.uk/sanger/Mouse_SnpViewer/rel-1410) [11, 20]. The SNP/Indel types selected for analysis were as follows: coding sequence variants, frame-shift variants, in-frame deletions, in-frame insertions, initiator codon variants, missense variants, regulatory region ablations, regulatory region amplifications, splice acceptor variants, splice donor variants, stop-gain variants, stop-loss variants, TFBS ablation, and TFBS amplification. The strains selected for analysis were CAST/EiJ and DBA/2J (C57BL6/J is the reference strain). The results were exported as a spreadsheet and sorted by the Snp/Indel consequence (Csq).

To further prioritize genes of interest within each of the Bayes credible intervals of the QTL, we determined which are expressed in eye tissue using the publicly available Gene Network database (www.genenetwork.org) [21]. The parameters used for the search were as follows: Species, Mouse (mm10); Group, BXD family; Type, Eye mRNA; Dataset, UTHSC BXD Young Adult Eye RNA-Seq (Nov20) TPM Log2. The "Get Any" field changed for each of the

QTL; each was searched based on the chromosome and Bayes credible interval, in megabases. The overlap of genes with at least one protein-altering DNA base-pair change and with ocular expression was compiled into a single spreadsheet each for each of the QTL (S1–S6 Tables).

## Results

### Phenotypes of parental strains, N2 mice, and F2 mice

The CAST and BXD24b parental mouse strains are genetically distinct from one another [22] and have overtly healthy eyes that differ significantly in ACD. We found that CAST mice have an ACD of 340.5 ± 20.7 μm, whereas BXD24b have an ACD of 360.5 ± 6.4 μm (9 and 12 mice, respectively; $p$ = 0.005). The phenotypic distributions of both the backcross (N2) and intercross (F2) mice followed a broad bell-shaped curve, suggesting the presence of many genes causing the difference in phenotype (S1 and S2 Figs). A Shapiro-Wilk goodness-of-fit test indicated that the data did not differ statistically from a normal distribution and, therefore, did not need to be transformed ($p$ = 0.88 and $p$ = 0.26, N2 and F2 respectively). ACD of the N2 progeny ranged from 319.8 μm to 412.4 μm (a difference of 92.6 μm; $n$ = 86 mice; S1 Fig) and the mean was 366.0 ± 20.4 μm; this is slightly thicker than the ACD of the parental strains. In the F2 progeny, the phenotypic distribution was like that in the N2 progeny (range = 296.8 μm to 409.0 μm; $n$ = 111 mice), and the mean ACD was within the range of the parental strains (344.7 ± 22.3 μm; S2 Fig).

An effective method of increasing the ability to detect and resolve QTL is by combining information from multiple mouse crosses [17, 23, 24]. This methodology was used on our datasets. The combined N2 and F2 dataset of ACD followed a normal distribution (goodness-of-fit, $p$ = 0.50; Fig 2), similar to the N2 and F2 crosses alone (S1 and S2 Figs).

### QTL analysis

Loci that affect the ACD phenotype were identified using N2 and F2 mice that were genotyped based on 93 polymorphic markers (see Fig 1); genotype: phenotype associations were assessed using R/qtl. A one-dimensional genome-wide scan of the N2 mice alone did not yield any statistically significant loci associated with the ACD phenotype but did show support for suggestive loci on chromosomes 4, 12, and 17 (Fig 3A). A one-dimensional scan of the F2 mice alone identified a significant locus on chromosome 11, and suggestive loci on chromosomes 2, 12, and 14 (Fig 3B). A one-dimensional scan of ACD across the combined dataset (86 N2 mice and 111 F2 mice) identified seven loci that passed the $p$ = 0.05 "significance" threshold: one on Chr 6, Chr 7, Chr 11, Chr 12, Chr 15, Chr 16, and Chr 17 (Fig 3C; dotted-dashed horizontal

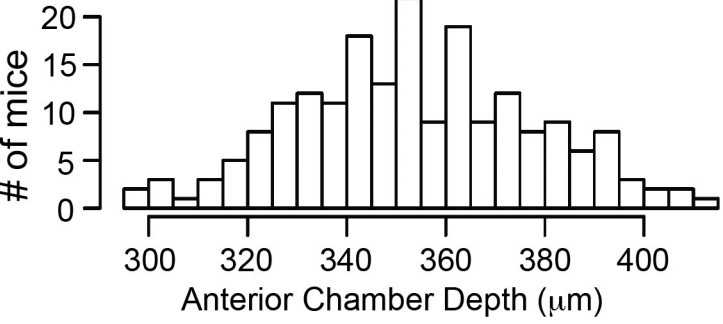

**Fig 2. Phenotypic distribution of anterior chamber depth for combined (BXD24b X CAST) N2 + F2 mice.**

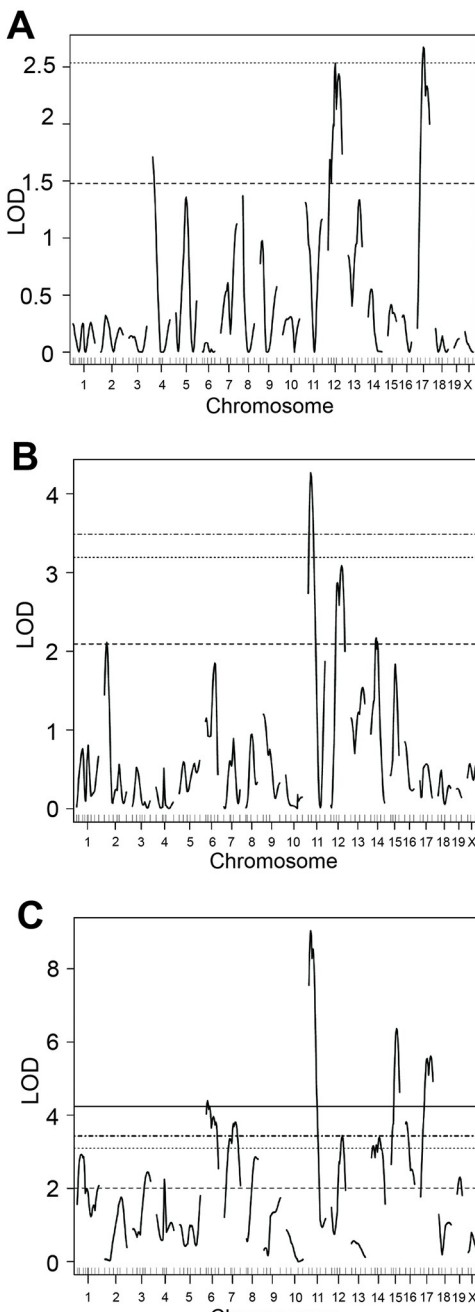

**Fig 3. One dimensional genome-wide scans (*i.e.*, scanone) with the anterior chamber depth phenotype.** (A) (BXD24b X CAST) N2 mice. Suggestive loci were observed on Chr 4, 12 and 17. (B) (BXD24b X CAST) F2 mice. A significant locus was observed on Chr 11; suggestive loci were observed on Chr 2, 12 and 14. (C) (BXD24b X CAST) N2 + F2 mice. Significant loci were observed on Chr 6, 7, 11, 12, 15, 16 and 17; suggestive loci were observed on Chr 1, 3, 4, 8, 14, and 19. Significance thresholds are as follows: solid line, $p = 0.01$; dashed-dotted, $p = 0.05$ (significant); dotted, $p = 0.1$; dashed, $p = 0.63$ (suggestive). LOD, logarithm of odds.

line). Suggestive loci were found on chromosomes 1, 3, 4, 8, 14, and 19 (Fig 3C; dashed horizontal line). A two-dimensional pairwise scan, which examines two loci simultaneously to consider epistatic interactions and/or additive effects, did not uncover any other statistically significant loci.

**Table 1. Multiple regression analysis.**

| QTL[a] | df[b] | Type III SS[c] | LOD[d] | %var[e] | F value[f] |
|---|---|---|---|---|---|
| 1@45.5 | 2 | 2052 | 1.939 | 1.836 | 3.942 |
| 3@26.0 | 2 | 3204 | 2.991 | 2.867 | 6.157 |
| 4@10.4 | 2 | 2254 | 2.126 | 2.017 | 4.332 |
| 6@5.8 | 2 | 4077 | 3.772 | 3.648 | 7.834 |
| 7@56.4 | 2 | 4156 | 3.841 | 3.718 | 7.985 |
| 8@57.9 | 2 | 1556 | 1.479 | 1.392 | 2.990 |
| 11@10.1 | 2 | 6379 | 5.762 | 5.707 | 12.257 |
| 12@31.0 | 2 | 10765 | 9.317 | 9.631 | 20.685 |
| 14@46.0 | 2 | 3397 | 3.165 | 3.039 | 6.527 |
| 15@29.3 | 2 | 7713 | 6.875 | 6.901 | 14.821 |
| 16@17.3 | 2 | 3030 | 2.834 | 2.711 | 5.821 |
| 17@27.5 | 2 | 4947 | 4.535 | 4.426 | 9.505 |
| 19@14.2 | 2 | 1358 | 1.293 | 1.215 | 2.609 |

Multiple regression analysis for the combined dataset (N2 + F2). Values shown are for the full model compared to a reduced model in which the indicated QTL is omitted.

[a] Chromosome and centimorgan position of the QTL

[b] Degrees of freedom

[c] Type III sum of squares

[d] Logarithm of the odds ratio

[e] Phenotypic variance (%) attributed to the indicated QTL

[f] F statistic

To verify the significance of the loci identified with the one-dimensional scans, multiple regression analysis was performed, comparing the full model to reduced models in which one locus is dropped at a time. For all analyses, the combined (N2 + F2) datasets were used. All loci exceeding the $p = 0.63$ suggestive significance threshold were included (Table 1), *i.e.*: a 13-QTL model was used. Support for a QTL was determined by comparing the LOD scores in Table 1 to the genome-wide significance thresholds. Multiple regression analysis of ACD resulted in evidence for QTL on chromosomes 6, 7, 11, 12, 15, and 17. Thus, this analysis provided additional evidence that at least 6 QTL regulate ACD in these mouse crosses. We have named these QTL as follows: Chr 6, *Acdq6* (anterior chamber depth QTL on Chr 6); Chr 7, *Acdq7*; Chr 11, *Acdq11*; Chr 12, *Acdq12*; Chr 15, *Acdq15*; and Chr 17, *Acdq17* (Table 2).

**Table 2. SNPs significantly associated with ACD.**

| QTL | Chr | SNP | Position[a] | LOD[b] |
|---|---|---|---|---|
| *Acdq6* | 6 | *rs3706286* | 12.14 | 4.16 |
| *Acdq7* | 7 | *rs3654689* | 61.81 | 3.70 |
| *Acdq11* | 11 | *rs3723833* | 19.21 | 8.29 |
| *Acdq12* | 12 | *rs3699929* | 36.28 | 2.69 |
| *Acdq15* | 15 | *rs3653368* | 25.09 | 6.25 |
| *Acdq17* | 17 | *rs3681815* | 39.02 | 5.10 |

[a] Position in centimorgans (cM)

[b] Logarithm of the odds ratio from scanone analysis

The SNPs with the maximum LOD score for each significant QTL were examined for their effects on the ACD phenotype in these mouse crosses (Table 2). At all loci except *Acdq12*, the allelic effects at the peak SNP revealed the CAST sequence promotes a decrease in anterior chamber depth (Fig 4). For all 5 of these loci, harboring BXD24b alleles appears to confer an additive increase in ACD. The allelic effect at the peak *Acdq12* SNP (*rs3699929*) suggests that

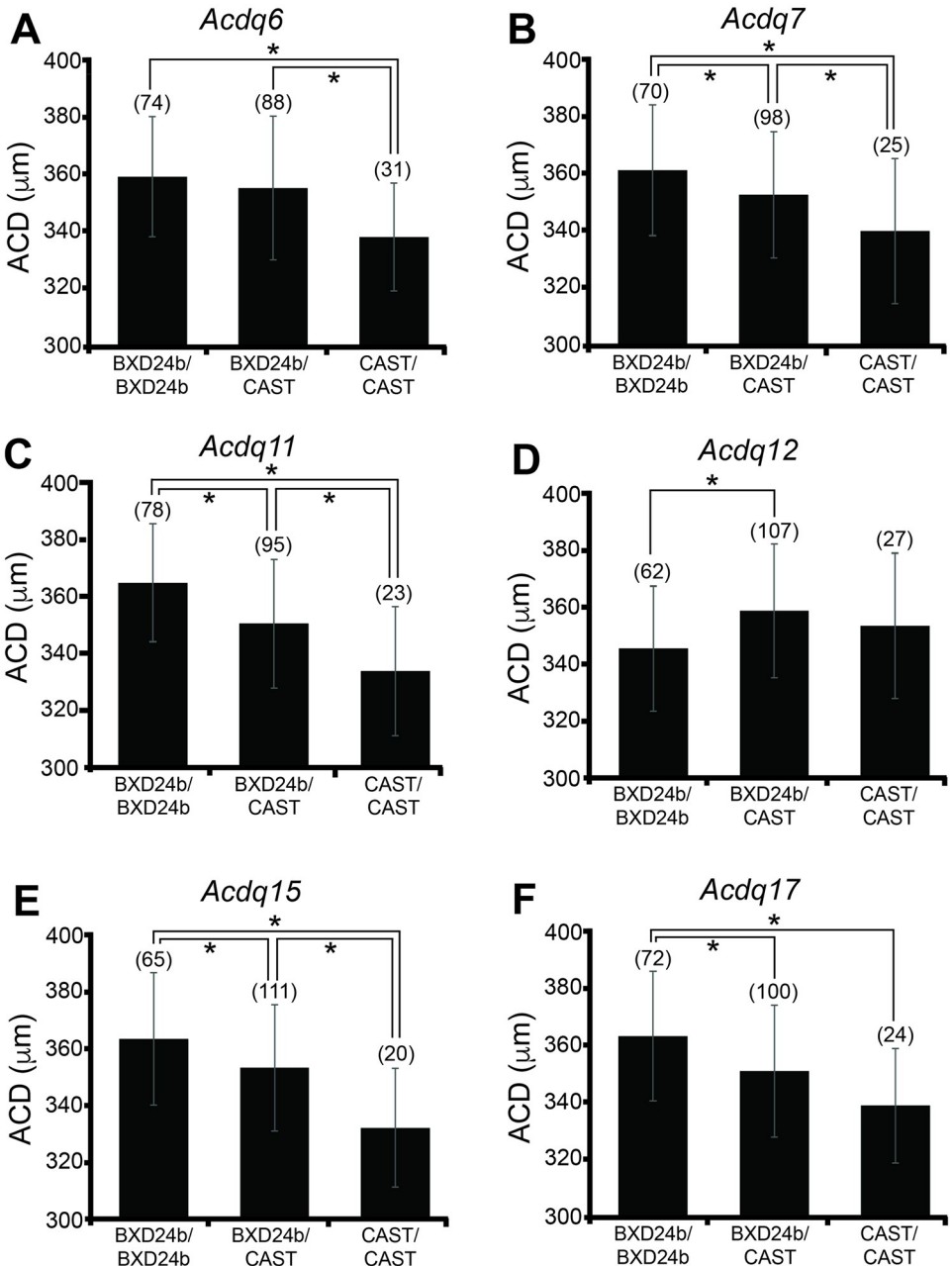

**Fig 4. Effect plots for combined (BXD24b X CAST) N2 + F2 mice.** Allelic effects of (A) *Acdq6* (Chr 6, *rs3706286*); (B) *Acdq7* (Chr 7, *rs3654689*); (C) *Acdq11* (Chr 11, *rs3723833*); (D) *Acdq12* (Chr 12, *rs3699929*); (E) *Acdq15* (Chr 15, *rs3653368*); (F) *Acdq17* (Chr 17, *rs3681815*) are included. The numbers in parentheses are the numbers of mice with the indicated genotype. The numbers do not add up to the same value for A-F because of missing genotypes at that SNP for a variable number of mice. Asterisks denote *p* < 0.05 significance. Error bars, standard deviation.

the CAST sequence might have a dominant effect at that locus and is associated with an increase in ACD (Fig 4).

## Bioinformatics analysis within the ACD-QTL loci

Genes within each QTL were prioritized based on two levels of filtering: first, by comparing known protein-coding and regulatory region differences between C57BL/6J or DBA/2J (the two strains from which BXD24b is derived) and CAST/EiJ, using the Wellcome Trust Sanger Institute's Mouse SNP query tool; second, by considering ocular gene expression using a publicly available database (www.genenetwork.org) [21]. Of the regulatory elements that were queried (regulatory region amplifications and ablations, transcription factor binding site amplifications and ablations) there were no known variants found for any of the loci. A summary of the number of genes with potential protein-altering variants is shown in Table 3. Detailed gene lists are provided as supporting information (S1–S6 Tables).

## Discussion

Examination of traits using a quantitative approach can be an effective and powerful means for studying medically important phenotypes [25, 26]. ACD is complex, highly heritable and follows a continuous distribution among the general population, with distinct variations across ethnicities [27]. Although ACD is an important ocular parameter, there is almost no knowledge about which genes influence the phenotype. In this study, we sought to identify genetic determinants of ACD by using mouse genetics to identify QTL. In crosses of BXD24b and CAST mice, significant ACD-modifying loci were identified on Chr 6 (*Acdq6*), Chr 7 (*Acdq7*), Chr 11 (*Acdq11*), Chr 12 (*Acdq12*), Chr 15 (*Acdq15*), and Chr 17 (*Acdq17*). Several loci were also found to pass the suggestive significance threshold in this analysis.

In humans, ACD changes during development of the eye, reaching its maximum distance once a person becomes a teenager. After that, ACD is relatively stable for one or two decades, after which, there is a slow decrease as a person ages [28, 29]. Short ACDs are associated with far-sightedness and an increased risk of primary angle closure glaucoma. The association with PACG is likely caused from the narrow angle that is formed where the iris and sclera meet circumferentially. Narrow angles result in reduced outflow and drainage through the trabecular meshwork and Schlemm's canal, leading to an increase in intraocular pressure (IOP) of the anterior chamber of the eye, a hallmark of PACG.

Considering the anatomical link between ACD and PACG, it seems likely that some shared genes control the phenotypes. Some of these genes will also likely regulate development of the

**Table 3. Bioinformatics summary data showing the number of genes within each QTL with potential protein-coding changes that are also expressed in mouse eyes.**

| QTL | Chr | missense | frameshift | start codon[a] | splice site[b] | stop codon[c] |
|-----|-----|----------|-----------|----------------|----------------|---------------|
| *Acdq6* | 6 | 526 | 33 | 3 | 37 | 29 |
| *Acdq7* | 7 | 547 | 25 | 7 | 35 | 35 |
| *Acdq11* | 11 | 77 | 7 | 1 | 14 | 2 |
| *Acdq12* | 12 | 187 | 7 | 5 | 19 | 14 |
| *Acdq15* | 15 | 68 | 2 | 1 | 10 | 3 |
| *Acdq17* | 17 | 132 | 4 | 0 | 11 | 5 |

[a] Includes any initiator codon variant

[b] Includes splice site acceptor and donor variants

[c] Includes stop codon lost and gained variants

anterior structures of the eye. To date, only one gene (*ABCC5*) has been reported to be associated with ACD and risk of PACG; it was identified with a GWA study in humans. Several genes, however, have been linked to PACG [30, 31]. A variety of protein functions are encoded by these genes, such as ocular development, ECM proteins or ECM modifying proteins, proteins involved in the cell cycle and apoptosis, and signaling proteins.

The same types of genes that contribute to PACG are in line with the types of genes we might expect to uncover for ACD. Gene products responsible for creating, maintaining, or remodeling ECM like the trabecular meshwork are reasonable candidates. Genes that define axial length (AL) could also contribute to ACD. Indeed, studies that have considered AL and ACD in tandem provide evidence that AL and ACD have shared genetic determinants [1]. One study demonstrated that of the 89% additive genetic factors for ACD, 25% were shared with those for AL [1]. Genes that define lens thickness could also play a role in ACD. A study by Lowe in 1970 reported that lens thickness and lens position affects the depth of the anterior chamber and the angle of the eye [5]. Similarly, genes that define iris phenotypes like thickness and curvature could also affect ACD. It is possible that genes responsible for corneal/scleral development might be also linked to ACD.

In our current study, we report 6 new genomic segments significantly associated with ACD. Although the gene lists for each QTL are too large for us to be confident at identifying causative alleles, these data provide concrete genomic locations that can be narrowed with future experiments (see S1–S6 Tables). Interestingly, three of our ACD-regulating QTL harbor genes previously identified to be linked to PACG: *Plekha7*, *Ltbp2*, and *Cyp1b1* [32–35]. All three of these genes are expressed in mouse eyes and harbor DNA changes between the strains used that lead to changes in the encoded proteins. *Plekha7*, a cell adhesion molecule, is in the interval for *Acdq7*. In our mice, there are two potential missense mutations within *Plekha7*. One is at *rs218172518*, changing from leucine in B6 mice to proline in CAST mice. The second possible mutation in *Plekha7* is at *rs255190982*, changing a valine in B6 mice to alanine in CAST mice (S2 Table). *Ltbp2* encodes a protein that helps organize and assemble ECM; it is found in the genomic interval for *Acdq12*. In our mice, there are up to 7 possible missense mutations and one potential in-frame deletion (S4 Table). In all instances, an amino acid would be changed relative to the B6 and CAST strains. *Cyp1b1*, which has been reported to affect IOP and organization of the trabecular meshwork, is within the *Acdq17* interval of our mouse strains. There is one possible missense mutation at *rs8255884* SNP changing a glycine in B6 to alanine in CAST, a relatively minor amino acid substitution (S6 Table).

Among the potential caveats of this study, three merit mentioning. The first pertains to the size of the QTL intervals. We chose to use an already established [12] panel of genome markers that were somewhat low-resolution. Because this was an experiment using controlled crosses with inbred mouse strains, we were limited by the number of recombination events within the mice and not on the number of markers (as opposed to what would be needed for a human GWAS). Thus, we thought it sufficient to use this panel of markers; indeed, we uncovered 6 statistically significant QTL this way. Recombination mapping of additional mice will need to be conducted to identify the causative alleles underlying each QTL. The bioinformatics analysis done here is merely a starting point for thinking about which genes might be contributing to the ACD phenotype. A second caveat of this study is our selection of animals used in the analysis. Because this is that an accompaniment to a study on retinal phenotypes, all animals included were homozygous for the *Cep290*[rd16] mutation. Any sequence variants linked to *Cep290* on chromosome 10 did not segregate in these crosses and our experimental design did not address any dependency on *Cep290*. Cep290 encodes a protein involved in cilia function and mutations within it are known to cause a variety of syndromic ciliopathy phenotypes. The *rd16* mutation harbored in our mice leads to an early onset retinal degeneration and

disfunction, as well as olfactory defects [36]. To our knowledge, there are no reports of *Cep290*$^{rd16}$ affecting the anterior chamber of the eye, so it seems unlikely that it would affect the ACD phenotype. In our examination of the mouse eyes, the anterior chambers appeared healthy, with apparent differences in thickness of the anterior chambers (this study) and the central corneas [12]. The third caveat pertains to the expression data acquired from the Gene Network database. It was collected from young adult eyes, long after eye development is complete. It would be more suitable to have used a dataset from very young animals with actively growing eyes; however, that data is not available to our knowledge.

## Conclusions

In sum, this study has identified a multi-genic pattern of ACD inheritance between two inbred strains of mice, and identifies six previously unrecognized loci, *Acdq6*, *Acdq7*, *Acdq11*, *Acdq12*, *Acdq15*, and *Acdq17*, of particular significance. These results are relevant not only to studies of ACD, but also to a broad array of ophthalmic studies using mice with the DBA/2J, C57BL/6J, or CAST/EiJ backgrounds—including the collaborative cross and Diversity Outbred mice [37–40], which would be under the influence of these same QTL.

## Supporting information

**S1 Fig. Phenotypic distribution of anterior chamber depth for (BXD24b X CAST) N2 mice.**
(PDF)

**S2 Fig. Phenotypic distribution of anterior chamber depth for (BXD24b X CAST) F2 mice.**
(PDF)

**S1 Table. Bioinformatics analysis of *Acdq6*.**
(XLSX)

**S2 Table. Bioinformatics analysis of *Acdq7*.**
(XLSX)

**S3 Table. Bioinformatics analysis of *Acdq11*.**
(XLSX)

**S4 Table. Bioinformatics analysis of *Acdq12*.**
(XLSX)

**S5 Table. Bioinformatics analysis of *Acdq15*.**
(XLSX)

**S6 Table. Bioinformatics analysis of *Acdq17*.**
(XLSX)

## Author Contributions

**Conceptualization:** Demelza R. Larson, Kacie J. Meyer, Michael G. Anderson.

**Data curation:** Demelza R. Larson, Allysa J. Kimber, Kacie J. Meyer.

**Formal analysis:** Demelza R. Larson, Allysa J. Kimber.

**Investigation:** Demelza R. Larson.

**Methodology:** Demelza R. Larson.

**Project administration:** Demelza R. Larson.

**Resources:** Michael G. Anderson.

**Supervision:** Demelza R. Larson.

**Writing – original draft:** Demelza R. Larson, Allysa J. Kimber.

**Writing – review & editing:** Demelza R. Larson, Kacie J. Meyer, Michael G. Anderson.

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
