## [Decision Letter · Decision Letter 0]

21 Jun 2023

PONE-D-23-15862Anterior chamber depth in mice is controlled by several quantitative trait lociPLOS ONE

Dear Dr. Larson,

Thank you for submitting your manuscript to PLOS ONE. The presented study is an original experimental research on a very relevant and interesting topic. After careful consideration, we feel that it has merit but does not fully meet PLOS ONE’s publication criteria as it currently stands. Therefore, we invite you to submit a revised version of the manuscript that addresses the points raised during the review process.

We look forward to receiving your revised manuscript.

Kind regards,

Fidan Aghayeva

Academic Editor

PLOS ONE

Journal Requirements:

2. To comply with PLOS ONE submissions requirements, in your Methods section, please provide additional information regarding the experiments involving animals and ensure you have included details on (1) methods of anesthesia and/or analgesia, and (2) efforts to alleviate suffering.

https://iovs.arvojournals.org/article.aspx?articleid=2430903

In your revision ensure you cite all your sources (including your own works), and quote or rephrase any duplicated text outside the methods section. Further consideration is dependent on these concerns being addressed.

Additional Editor Comments:

The reviewers raised significant concerns and AE agrees to them. I would like you to address these issues before reaching to conclusion for publication. Please pay careful attention to all concerns raised by two reviewers. It is likely that your revised manuscript will be returned to at least one more referee. Sometimes, an expert who was not part of the initial review process will also be invited to comment on the revision. Criticisms that were not mentioned during the initial review may arise at a future stage of the peer review process. Please pay careful attention to points raised by two reviewers.

Reviewers' comments:

Reviewer's Responses to Questions

**Comments to the Author**

1. Is the manuscript technically sound, and do the data support the conclusions?

Reviewer #1: Yes

Reviewer #2: Partly

2. Has the statistical analysis been performed appropriately and rigorously? 

Reviewer #1: Yes

Reviewer #2: Yes

3. Have the authors made all data underlying the findings in their manuscript fully available?

Reviewer #1: Yes

Reviewer #2: Yes

4. Is the manuscript presented in an intelligible fashion and written in standard English?

PLOS ONE does not copy edit accepted manuscripts, so the language in submitted articles must be clear, correct, and unambiguous. Any typographical or grammatical errors should be corrected at revision, so please note any specific errors here.

Reviewer #1: Yes

Reviewer #2: Yes

5. Review Comments to the Author

Reviewer #1: The authors demonstrate a quantitative multi-genic pattern of ACD inheritance in mice and identify six previously unrecognized ACD-influencing loci. Interestingly, three of ACD-regulating QTL harbor genes previously identified to be linked to PACG: Plekha7, Ltbp2 and Cyp1b1.The study is focused on a meaningful topic and deserves consideration. However, some points should be clarified.

1. The authors did not explain clearly why BXD24b and CAST mice were chosen for research. The potential correlation between different strains and anatomical parameters of the eyeball should be clarified in the discussion.

2. line 311-312 “our dataset showed no correlation between central corneal thickness (CCT) and ACD (data not shown).” conclusion without data support can be removed.

3. The gene lists for each QTL are too large, the physical sizes of the QTL need to be narrowed through recombination mapping in the future. As the authors mentioned in the manuscript, these results are relevant not only to studies of ACD, but also to a broad array of ophthalmic studies. If the authors can analyze multiple anatomical parameters of the eyeball simultaneously, the results will be more directional and valuable for reference.

4. Analysis of ACD was an accompaniment to an ongoing study using the same strains of mice to examine retinal phenotypes associated with the Cep290 gene, and all mice studied were homozygous for the Cep290rd16 mutation. It is known that over 100 unique CEP290 mutations have been identified, and no clear genotype-phenotype correlations could yet be established. [PMID: 20690115]. The authors should explain the phenotypes that Cep290 mutations may cause and whether they may have an impact on ACD in the discussion section.

Reviewer #2: In this study, Larson et al. utilized the BXD24b and CAST strains to establish a QTL mapping population through hybridization and backcrossing strategies. They employed 93 genome-wide SNPs as markers to conduct QTL mapping and investigated the relationship between anterior chamber depth (ACD) in the animal population and the identified QTLs. Remarkably, the authors successfully identified six novel QTLs that exert a significant impact on ACD

Major comments:

1. Currently, there are multiple high-precision methods available for QTL mapping. However, the authors chose a lower-resolution approach as described in the manuscript. This decision significantly compromises the accuracy and richness of the obtained results.

2. The animal population used in this study is derived from the authors' previous research on CCT. Although the authors emphasize that the use of this population was motivated by animal welfare considerations, the presence of the CEP290 homozygous mutation in this population introduces unpredictability that adds uncertainty to the results of this study. Furthermore, in order to further validate the findings of this study, it would be necessary to construct a new mapping population, which, from this perspective, does not necessarily enhance animal welfare.

3. While identifying three potential genes is noteworthy, without any validation, the contribution to the current knowledge base is limited. It is important to perform additional experiments to validate the findings.

Minor comments:

The authors utilized QTL mapping data from their previous publication in IOVS 2015 (https://doi.org/10.1167/iovs.15-17179). Although new phenotypic measurements and data analysis were conducted, Figure 1 appears to be the same as in the previous publication, without proper labeling or citation. It is recommended to optimize and improve Figure 1 by incorporating the new results obtained from this study.

6. PLOS authors have the option to publish the peer review history of their article (what does this mean?). If published, this will include your full peer review and any attached files.

Reviewer #1: No

Reviewer #2: No

---

## [Author Response · Author response to Decision Letter 0]

1 Aug 2023

AE comments:

We have examined the formatting and file naming requirements of PLOS ONE and have made the appropriate changes.

2. Please provide additional information regarding the experiments involving animals and ensure you have included details on (1) methods of anesthesia and/or analgesia, and (2) efforts to alleviate suffering. 

We have examined the Materials and Methods section and have the information regarding anesthesia/analgesia and reduction of animal suffering under the subheading of “ACD phenotyping”. We included additional text describing the non-invasive and painless nature of the phenotyping procedure. The relevant text is on lines 148 – 156 of the revised manuscript.

3. We noticed you have some minor occurrence of overlapping text with the following previous publication(s), which needs to be addressed. 

We have examined the locations of overlapping text and have modified the language accordingly.

4. We note that you have included the phrase “data not shown” in your manuscript…if the data are not a core part of the research being presented in your study, we ask that you remove the phrase that refers to these data. 

The sentence with the phrase “data not shown” has been modified (line 485).

5. Please include captions for your Supporting Information files at the end of your manuscript. 

Captions have been included for the supporting information files.

Reviewer 1:

1. The authors did not explain clearly why BXD24b and CAST mice were chosen for research. The potential correlation between different strains and anatomical parameters of the eyeball should be clarified in the discussion. 

A brief explanation of why the BXD24b and CAST mouse strains were used is on lines 114 – 117 within the Materials and Methods of the manuscript. A second explanation for why these strains were used is given on lines 518 – 522 of the Discussion. A brief explanation of ocular health and differences in ACD between the two strains is given on lines 245 – 248 within the Results section. Additionally, our approach taken was guided by what was financially achievable. While the experiment had caveats, it none-the-less had meaningful outcomes. We have attempted to keep the discussion focused to the points of greatest interest to the readership and would prefer to do so in the revised manuscript as well. We feel any additional discussion of the background strains used would be distracting. 

2. line 311-312 “our dataset showed no correlation between central corneal thickness (CCT) and ACD (data not shown).” conclusion without data support can be removed. 

The sentence that included the phrase “data not shown” has been modified (line 485).

3. The gene lists for each QTL are too large, the physical sizes of the QTL need to be narrowed through recombination mapping in the future. 

We agree that the QTL are too large to attempt to name causative alleles and that recombination mapping would need to be conducted in the future (this is listed as one of the caveats of the study on lines 509 – 518 of the Discussion). Given the amount of time it takes to successfully identify causative alleles of QTL using recombination mapping in mice, we thought it prudent to save that for a future study so that the current data could be quickly disseminated to the community. It is also relevant that this work was done in the context of a small liberal arts college primarily serving undergraduates and we have no feasible means of obtaining the budget that would be required for additional fine-scale mapping. Thus, the time and budget for this both make it beyond our current scope. In sharing our current results, we hope that either we, or other groups seeing the data, will be able to build on the project to better consider candidates in future work.

…If the authors can analyze multiple anatomical parameters of the eyeball simultaneously, the results will be more directional and valuable for reference. 

This cohort of mice is being (and has been) used to examine 3 ocular parameters: retinal thickness (and likewise, retinal degeneration; a forthcoming manuscript), central corneal thickness (CCT; a published article) and ACD (current manuscript). The study on retinal thickness is ongoing and will be published; this is implied in the current manuscript in both the Materials and Methods (lines 114 – 117) and the Discussion (line 519). The CCT paper was referenced several times throughout this manuscript.

4. …The authors should explain the phenotypes that Cep290 mutations may cause and whether they may have an impact on ACD in the discussion section. 

We agree and we note all our mice are homozygous for the Cep290rd16 mutation and this causes a limitation of the study. We have modified the language about Cep290 in the Discussion to briefly include some of the phenotypes its mutant form can cause. However, we feel an extensive discussion of this will detract from the main points of the paper. We believe the above modifications address this point fairly and that any further elaboration of the effects of Cep290rd16 on ACD would become speculative. Nevertheless, we are happy to do so if advised by the Reviewer and/or Editor. 

Reviewer 2:

1. Currently, there are multiple high-precision methods available for QTL mapping. However, the authors chose a lower-resolution approach as described in the manuscript. This decision significantly compromises the accuracy and richness of the obtained results. 

It is true that we chose to use an already-published, low-resolution approach to the QTL mapping of this study. The rationale behind this was two-fold. First and foremost, because the study employed controlled crosses with inbred mouse strains, narrowing of the QTL intervals is limited by the number of naturally occurring recombination events among markers rather than the number of markers themselves (this information was added to the Discussion, lines 512 – 514). Thus, while tiling in more markers for these mice may narrow the boundaries some, the intervals will remain large. This contrasts with genome-wide association studies in humans; since those populations are outbred and random, many genetic markers are required. The use of inbred mouse strains simplifies the experiment and warrants the use of a relatively small number of genetic markers to identify relevant, and significantly associated, loci. Second, the current study was conducted at a small, primarily undergraduate institution with very little funding for research. It was fiscally practical for us to use the genotype data that was already in place for these mice, especially since using a different marker set was not likely to change the results in a substantiative way. 

2. The animal population used in this study is derived from the authors' previous research on CCT…the presence of the CEP290 homozygous mutation in this population introduces unpredictability that adds uncertainty to the results of this study…it would be necessary to construct a new mapping population, which, from this perspective, does not necessarily enhance animal welfare. 

Our motivation for reusing the animal population that is currently being used for the retinal degeneration study (a forthcoming manuscript) and was used for the CCT study, was really to capture as much phenotypic information from these animals as possible. Maximizing the use of animals is both ethical for the animals and fiscally responsible. It is true that we will need another set of animals to perform the recombination mapping, but that would be the case if we used a unique set of animals for this study or reused animals from a concurrent study. This way, we did reduce the number of animal lives that were/will be sacrificed.

3. While identifying three potential genes is noteworthy, without any validation, the contribution to the current knowledge base is limited. It is important to perform additional experiments to validate the findings. 

We completely agree that at this time, causative genes cannot be named as underlying any of the QTL. We emphasized that limitation in the Discussion (lines 515 – 518) and only mentioned the 3 genes we did as a point of intrigue. These future validation experiments will be useful, but we feel they are not necessary for the current study. We believe it is important to quickly disseminate our current findings, rather than wait on recombination mapping experiments, which can take years. In sharing our current results, we hope that either we, or other groups seeing the data, will be able to build on the project to better consider candidates in future work.

Minor comment: 

The authors utilized QTL mapping data from their previous publication in IOVS 2015 (https://doi.org/10.1167/iovs.15-17179). Although new phenotypic measurements and data analysis were conducted, Figure 1 appears to be the same as in the previous publication, without 

proper labeling or citation. It is recommended to optimize and improve Figure 1 by incorporating the new results obtained from this study. 

We did remake Figure 1 from scratch, and it is not identical to the one found in the CCT paper. The CCT paper used 96 genetic markers; this study used 93. In this current ACD study, Chr 4, Chr 8, and Chr 9 each have 1 less marker than was used in the CCT study, although all the other genetic markers used were identical. The two figures look so similar because the output of the R/qtl program (where the figure was produced) creates one type of picture for genetic marker data. Nevertheless, we have added a citation to the Figure 1 caption that references our earlier CCT paper.

---

## [Decision Letter · Decision Letter 1]

16 Aug 2023

Anterior chamber depth in mice is controlled by several quantitative trait loci

PONE-D-23-15862R1

Dear Dr. Demelza Larson, Ph.D,

We’re pleased to inform you that your manuscript has been judged scientifically suitable for publication and will be formally accepted for publication once it meets all outstanding technical requirements.

Kind regards,

Fidan Aghayeva

Academic Editor

PLOS ONE

Additional Editor Comments (optional):

Reviewers' comments:

Reviewer's Responses to Questions

**Comments to the Author**

1. If the authors have adequately addressed your comments raised in a previous round of review and you feel that this manuscript is now acceptable for publication, you may indicate that here to bypass the “Comments to the Author” section, enter your conflict of interest statement in the “Confidential to Editor” section, and submit your "Accept" recommendation.

Reviewer #3: All comments have been addressed

2. Is the manuscript technically sound, and do the data support the conclusions?

Reviewer #3: Yes

3. Has the statistical analysis been performed appropriately and rigorously? 

Reviewer #3: Yes

4. Have the authors made all data underlying the findings in their manuscript fully available?

Reviewer #3: Yes

5. Is the manuscript presented in an intelligible fashion and written in standard English?

Reviewer #3: Yes

6. Review Comments to the Author

Reviewer #3: In the manuscript entitled “Anterior chamber depth in mice is controlled by several quantitative trait loci” have used mice to determine the genetics of ACD using a backcross and F1 intercross strategy coupled with QTL analysis. The authors have answered all the questions posed by the reviewers in a satisfactory manner.

This reviewer has only one question that can be addressed in the discussion prior to publication- do the BXD24/TyJ mice and the N2 and F2 crosses display pigment dispersion at the age of ACD measurement. Could this be a confounding factor and mask phenotype changes?

7. PLOS authors have the option to publish the peer review history of their article (what does this mean?). If published, this will include your full peer review and any attached files.

Reviewer #3: No

---

## [Editor Report · Acceptance letter]

18 Aug 2023

PONE-D-23-15862R1 

Anterior chamber depth in mice is controlled by several quantitative trait loci 

Dear Dr. Larson:

I'm pleased to inform you that your manuscript has been deemed suitable for publication in PLOS ONE. Congratulations! Your manuscript is now with our production department. 

Kind regards, 

on behalf of

Dr. Fidan Aghayeva 

Academic Editor

PLOS ONE